# Applicability and cost-effectiveness of the Systolic Blood Pressure Intervention Trial (SPRINT) in the Chinese population: A cost-effectiveness modeling study

Chao Li[1,2], Kangyu Chen[3], Victoria Cornelius[4], Ewan Tomeny[5], Yang Wang[6], Xiaowei Yang[7], Xiaodan Yuan[8], Rui Qin[8], Dahai Yu[9], Zhenqiang Wu[10], Duolao Wang[2], Tao Chen[1,2]*

1 Department of Epidemiology and Health Statistics, School of Public Health, Xi'an Jiaotong University Health Science Center, Xi'an, China, 2 Global Health Trials Unit, Department of Clinical Sciences, Liverpool School of Tropical Medicine, Liverpool, United Kingdom, 3 Department of Cardiology, First Affiliated Hospital of USTC, Division of Life Sciences and Medicine, University of Science and Technology of China, Hefei, China, 4 Imperial Clinical Trials Unit, School of Public Health, Imperial College London, London, United Kingdom, 5 Centre for Applied Health Research & Delivery, Liverpool School of Tropical Medicine, Liverpool, United Kingdom, 6 Medical Research and Biometrics Centre, Fuwai Hospital, National Centre for Cardiovascular Disease, Peking Union Medical College and Chinese Academy of Medical Sciences, Beijing, China, 7 School of Public Policy and Administration, Xi'an Jiaotong University, Xi'an, China, 8 Department of Health Education, Jiangsu Province Hospital of Integration of Chinese and Western Medicine, Nanjing, China, 9 Arthritis Research UK Primary Care Centre, Research Institute for Primary Care and Health Sciences, Keele University, Keele, United Kingdom, 10 Department of Geriatric Medicine, University of Auckland, Auckland, New Zealand

* Tao.chen@lstmed.ac.uk

**Data Availability Statement:** CHARLS data are available via the website: http://charls.pku.edu.cn/pages/data/2011-charls-wave1/en.html Other relevant data are within the manuscript and its Supporting information files.

## Abstract

### Background

The Systolic Blood Pressure Intervention Trial (SPRINT) showed significant reductions in death and cardiovascular disease (CVD) risk with a systolic blood pressure (SBP) goal of <120 mm Hg compared with a SBP goal of <140 mm Hg. Our study aimed to assess the applicability of SPRINT to Chinese adults. Additionally, we sought to predict the medical and economic implications of this intensive SBP treatment among those meeting SPRINT eligibility.

### Methods and findings

We used nationally representative baseline data from the China Health and Retirement Longitudinal Study (CHARLS) (2011–2012) to estimate the prevalence and number of Chinese adults aged 45 years and older who meet SPRINT criteria. A validated microsimulation model was employed to project costs, clinical outcomes, and quality-adjusted life-years (QALYs) among SPRINT-eligible adults, under 2 alternative treatment strategies (SBP goal of <120 mm Hg [intensive treatment] and SBP goal of <140 mm Hg [standard treatment]). Overall, 22.2% met the SPRINT criteria, representing 116.2 (95% CI 107.5 to 124.8) million people in China. Of these, 66.4%, representing 77.2 (95% CI 69.3 to 85.0)

**Funding:** The authors received no specific funding for this work.

**Competing interests:** The authors have declared that no competing interests exist.

**Abbreviations:** BP, blood pressure; CHARLS, China Health and Retirement Longitudinal Study; CVD, cardiovascular disease; HF, heart failure; ICER, incremental cost-effectiveness ratio; MI, myocardial infarction; QALY, quality-adjusted life-year; SAE, serious adverse event; SBP, systolic blood pressure; SPRINT, Systolic Blood Pressure Intervention Tria.

million, were not being treated for hypertension, and 22.9%, representing 26.6 (95% CI 22.4 to 30.7) million, had a SBP between 130 and 139 mm Hg, yet were not taking antihypertensive medication. We estimated that over 5 years, compared to standard treatment, intensive treatment would reduce heart failure incidence by 0.84 (95% CI 0.42 to 1.25) million cases, reduce CVD deaths by 2.03 (95% CI 1.44 to 2.63) million cases, and save 3.84 (95% CI 1.53 to 6.34) million life-years. Estimated reductions of 0.069 (95% CI −0.28, 0.42) million myocardial infarction cases and 0.36 (95% CI −0.10, 0.82) million stroke cases were not statistically significant. Furthermore, over a lifetime, moving from standard to intensive treatment increased the mean QALYs from 9.51 to 9.87 (an increment of 0.38 [95% CI 0.13 to 0.71]), at a cost of Int$10,997 per QALY gained. Of all 1-way sensitivity analyses, high antihypertensive drug cost and lower treatment efficacy for CVD death resulted in the 2 most unfavorable results (Int$25,291 and Int$18,995 per QALY were gained, respectively). Simulation results indicated that intensive treatment could be cost-effective (82.8% probability of being below the willingness-to-pay threshold of Int$16,782 [1× GDP per capita in China in 2017]), with a lower probability in people with SBP 130–139 mm Hg (72.9%) but a higher probability among females (91.2%). Main limitations include lack of specific SPRINT eligibility information in the CHARLS survey, uncertainty about the implications of different blood pressure measurement techniques, the use of several sources of data with large reliance on findings from SPPRINT, limited information about the serious adverse event rate, and lack of information and evidence for medication effectiveness on renal disease.

## Conclusions

Although adoption of the SPRINT treatment strategy would increase the number of Chinese adults requiring SBP treatment intensification, this approach has the potential to prevent CVD events, to produce gains in life-years, and to be cost-effective under common thresholds.

## Author summary

### Why was this study done?

- The Systolic Blood Pressure Intervention Trial (SPRINT) has previously demonstrated significant reductions in death and cardiovascular disease (CVD) risk with a systolic blood pressure (SBP) goal of < 120 mm Hg (intensive treatment) compared with a SBP goal of <140 mm Hg (standard treatment).

- A large proportion of Chinese adults are classified as hypertensive but few of them achieve recommended blood pressure targets.

### What did the researchers do and find?

- We used nationally representative data from the China Health and Retirement Longitudinal Study (CHARLS) (2011–2012) to assess the applicability of SPRINT to the Chinese

adult population, and a validated microsimulation model to predict the cost-effectiveness of this intensive treatment among those meeting the SPRINT eligibility criteria for intensive treatment.

- It is estimated that 116.2 million adults in China are eligible for the SPRINT intensive treatment strategy.

- If adopted, intensive treatment has the potential to prevent 2.03 million CVD deaths, with a potential gain of 3.84 million more life-years over 5 years, and would likely be cost-effective over a lifetime.

## What do these findings mean?

- A substantial number of Chinese adults meet SPRINT eligibility criteria for intensive blood pressure treatment.

- This evidence suggests that intensive treatment among high-risk populations could be cost-effective.

## Introduction

Elevated blood pressure (BP) is the leading risk factor for cardiovascular diseases (CVDs) in China [1]. In 2015 a systolic BP (SBP) of 140 mm Hg or higher was found to be associated with over 1.7 million deaths and more than 32 million disability-adjusted life-years in China, accounting for more than 22% of global health losses from hypertension over that year [2]. Despite safe and effective antihypertensive medications having been available for decades and recommended for use under the current guideline [3,4], China still faces huge challenges with its low hypertension control rate and increasing hypertension in both prevalence and absolute numbers [5–7]. It is clear that better management is required to address this growing burden of hypertension.

The landmark Systolic Blood Pressure Intervention Trial (SPRINT) found that participants assigned to intensive treatment (SBP goal of <120 mm Hg) had an 25% reduction in major cardiovascular events and deaths compared with standard treatment (SBP goal of <140 mm Hg), and all-cause mortality was reduced by 27% [8]. This significant finding was supported by 2 meta-analyses of randomized clinical trials [9,10] and had far-reaching implications for hypertension guidelines across the world [11,12].

The generalizability of the SPRINT intensive SBP treatment strategy to the Chinese population is unclear, and the costs incurred from intensive drug treatment and adverse events need to be weighed against the gains from preventing cardiovascular morbidity and mortality. We explored the potential benefits that could be achieved through implementing the intensive SBP treatment goal strategy proposed in SPRINT among Chinese adults. Our study used a nationally representative dataset from China and a microsimulation model, aiming to (1) estimate the numbers of Chinese adults who meet the SPRINT eligibility criteria and describe their characteristics, (2) predict medium- and long-term associated cardiovascular risk and mortality, and (3) evaluate the lifetime cost-effectiveness of intensive SBP treatment versus standard treatment among adults meeting SPRINT eligibility criteria.

## Methods

This study is composed of 2 parts: population analysis and health economic modeling. It was planned at the time of study conception on the basis of our previous work [13], although no formal prospective analysis plan was recorded. A further analysis of splitting the total cost into its components was added in response to a peer reviewer's comments (see Table G in S1 Text). This study is reported as per the Consolidated Health Economic Evaluation Reporting Standards (CHEERS) guideline (S1 CHEERS Checklist).

### Design of population study

**Data source and study population.** We used the national baseline database of the China Health and Retirement Longitudinal Study (CHARLS), which was conducted in 2011–2012 and included 17,708 people. The study design for CHARLS has been reported previously [14,15]. CHARLS is a nationally representative survey of people aged 45 years and older in China, and participants are followed up biennially. CHARLS data are available via the website http://charls.pku.edu.cn/pages/data/2011-charls-wave1/en.html. Other relevant data are presented here and in S1 Text. Study participants are selected through multistage probability sampling and can be weighted to obtain national estimates. Detailed data including demographic characteristics, medical history, prescription drug use, clinical measurement, and laboratory testing are collected by standard questionnaires. Definitions for variables used in our study can be found in "Study variables and definition" in S1 Text. The ethics committee of Peking University Health Science Center approved CHARLS, and all participants gave written informed consent before participation.

**Analytical approach.** Potential eligibility for SPRINT was determined using a multistep algorithm. The sample needed to meet the following criteria: age $\geq$ 50 years; SBP 130–180 mm Hg on 0 or 1 antihypertensive medication class, 130–170 mm Hg on up to 2 classes, 130–160 mm Hg on up to 3 classes, or 130–150 mm Hg on up to 4 classes; and the presence of 1 or more high CVD risk conditions. Despite being part of the eligibility criteria, antihypertensive medication classes are not reported in the CHARLS database, and so in the present study we used the slightly less restrictive criterion SBP 130–180 mm Hg on any number of antihypertensive medication classes. High CVD risk conditions were defined as 1 or more of the following: history of coronary heart disease, estimated glomerular filtration rate (eGFR) of 20 to 59 ml/min/1.73 m$^2$, 10-year risk for CVD $\geq$ 15% calculated using the Framingham risk score for general clinical practice, and age $\geq$ 75 years. Participants were not eligible if they had any of the following exclusion criteria: diabetes, a history of stroke, heart failure (HF), or end-stage renal disease. The demographic and clinical characteristics of Chinese adults meeting SPRINT eligibility criteria are presented in Table A in S1 Text. The absolute numbers overall and for subgroups of Chinese adults meeting each sequential SPRINT eligibility criterion were estimated using survey analyses with patient-level sampling weights for China.

### Design of the health economic model

We developed a microsimulation model to project healthcare costs, clinical outcomes, and quality-adjusted life-years (QALYs) among the SPRINT-eligible Chinese adults for antihypertensive treatment (regardless of their BP treatment history) under 2 alternative SBP treatment strategies: SBP goal of <120 mm Hg (intensive treatment) versus SBP goal of <140 mm Hg (standard treatment). In order to maintain the correlations between the attributes of individuals (e.g., age, sex, SBP), we bootstrapped 10,000 real individuals meeting SPRINT eligibility criteria from CHARLS using TreeAge Pro 2018 R1.1. This approach contrasts with simulating

hypothetical individuals via the distribution of each independent variable (e.g., mean and standard deviation for age), as the authors have done previously [13].

**Model structure.**   In line with our previous work [13], our model is structured around 9 distinct health states. At the end of every model cycle (i.e., 1 year), an individual will either transit to another health state or remain in the same state, with the probabilities sampled from defined distributions. Subsequent to a primary event (i.e., myocardial infarction [MI], HF, or stroke), patients move to a chronic heath state in which they have a higher risk of experiencing a further CVD event or death. Also, our model includes the disease pathway of experiencing more than 1 event (e.g., the occurrence of stroke after MI) during the first year of primary event. In SPRINT, it was reported that intensive SBP treatment could increase the risk of specific adverse events, such as hypotension, syncope, electrolyte abnormalities, and acute kidney injury. However, due to the lack of local data from China, we did not specify the individual serious adverse events (SAEs) as in SPRINT. Therefore, we included a category of combined SAEs to capture the costs and harms incurred from intensive SBP treatment. Additionally, we assumed the risk of a CVD event would be the same for those with and without SAEs in our model. The overall disease path is described in S1 Fig, and the detailed disease states within each path can be found in our previous publication [13].

**Transition probabilities.**   We estimated the primary incidence of MI/stroke [16] and HF [17] by their corresponding and validated risk equations using the individual characteristics of the SPRINT-eligible people in CHARLS. Other probabilities—including those of chronic conditions followed by a primary event, secondary events, and SAEs from antihypertensive medications—were derived from national sources and published literature. The values for the transition probabilities and their distributions are summarized in Table C in S1 Text.

**Cost and utility weights.**   Costs were considered from a payer's perspective. All costs are reported in 2017 international dollars according to the exchange rate published by the World Bank, based on purchasing power parity (PPP) methods (Int$1.00 = 3.54 yuan) [18]. All future costs and QALYs were discounted at 3% annually. Total costs included antihypertension treatment costs, cost of complications, and SAE costs. Specifically, for costs of BP medications, we used a nationwide cross-sectional survey on the costs and frequencies of each standard antihypertensive drug class within the combinations of medications [4] recommended by 2018 Chinese guidelines for the management of hypertension [19]. For the costs of a CVD event, we incorporated a hospitalization cost associated with such an acute event, as well as annual costs afterwards. The cost of SAEs was from a publication [20] but was only applied in the year when an SAE occurred. A detailed description of costs can be found in the "Cost" section of S1 Text and in Table D in S1 Text.

We calculated utilities by age and sex for patients with hypertension based on a Chinese population; however, as in our previous work [13], for a CVD event we adopted utility values used in other cost-effectiveness studies. Since utilities associated with SAEs were not available, we assumed a utility of 0.5 for 1 week before returning to the previous health state. This approach has been adopted and justified in previous studies [21,22].

**Model validation.**   We validated the model through 2 strategies: First, we compared the projected life expectancy in our model with that from China life tables in 2017 [6]. Second, we also computed the predicted 5-year coronary heart disease, stroke, HF, and mortality rates among people with standard treatment from our base-case model and compared these with published data.

**Base-case and sensitivity analyses.**   Since there is no commonly accepted cost-effectiveness threshold for China, we adopted the one calculated using the method previously recommended by the World Health Organization [23]. Here, an incremental cost-effectiveness ratio

(ICER) below Int$16,782 (China's gross domestic product per capita in 2017) implies that the intensive SBP treatment is cost-effective compared to standard treatment.

In order to examine the effect of the uncertainty of each model parameter on the result from our base-case analysis, we varied each input value in our model over a plausible range in a series of 1-way sensitivity analyses. In addition to the above deterministic sensitivity analyses, we conducted probabilistic sensitivity analyses by running our model 1,000 times, with each run sampling from prespecified distributions of each parameter. The plausible ranges and distributions for each of the model parameters are detailed in Table C and Table D in S1 Text.

## Results

### Number and characteristics of Chinese adults meeting SPRINT eligibility criteria

Overall, 22.2% (95% CI 20.5% to 23.9%) of Chinese adults, representing 116.2 (95% CI 107.5 to 124.8) million people, met the SPRINT criteria (Figs 1 and 2). Among Chinese adults with hypertension, 41.4% (95% CI 37.8% to 45.1%), or 93.3 million, were eligible for SPRINT. This is compared to 37.9% (95% CI 34.4%, 41.5%), or 22.8 million, among Chinese adults without hypertension (i.e., with SBP 130–139 mm Hg). Additionally, among those with treated hypertension, 37.2% (95% CI 33.6% to 40.8%), or 39.0 million, may benefit from further SBP reduction (Fig 2). The number of adults meeting each sequential SPRINT eligibility criterion are detailed in Table B in S1 Text.

Compared with the overall study population, those eligible for SPRINT were more likely to be older, to be male, to smoke, and to have a SBP $\geq$ 140 mm Hg (Table A in S1 Text). It is noted that, of those SPRINT-eligible adults, 77.2 million (66.4%) were not being treated for hypertension, and 26.6 million (22.9%) had a SBP of 130 to 139 mm Hg, yet were not taking antihypertensive medications.

### Model validation

The predictions of our microsimulation model for life expectancy and rates of MI, stroke, CVD mortality, and all-cause mortality over the 5-year period were similar to those published in other real-world Chinese studies (Table E in S1 Text).

### Base-case analysis

Compared to standard treatment, intensive SBP treatment was predicted to reduce the incidence rate of HF within 5 years by 0.72% (95% CI 0.36%, 1.08%) and CVD mortality by 1.75% (95% CI 1.24%, 2.26%), though it was not predicted to significantly reduce the incidence rate for MI (0.06%; 95% CI −0.24, 0.36%) or stroke (0.31%; 95% CI 0.09%, 0.71%). Correspondingly, we estimate the 5-year event reduction would be 0.84 (95% CI 0.42, 1.25) million for HF, 2.03 (95% CI 1.44, 2.63) million for CVD mortality, 0.069 (95% CI −0.28, 0.42) million for MI, and 0.36 (95% CI −0.10, 0.82) million for stroke. Taken together, 3.84 (95% CI 1.53, 6.34) million life-years could be saved within 5 years via intensive SBP treatment compared to standard treatment, and 11.06 (95% CI 9.7, 14.73) million life-years over a lifetime (Fig 3; Table F in S1 Text).

The potential changes in QALYs and total cost (consisting of antihypertension treatment costs, cost of SAEs, and cost of complications) over a lifetime are summarized in Table 1 and Table G in S1 Text. Intensive SBP treatment led to a gain of 0.38 QALYs (95% CI 0.13, 0.71), while increasing costs from $7,861 to $11,395 (an increment of $3,777 [95% CI −$208, $8,286]). This means that compared to standard treatment, intensive SBP treatment costs

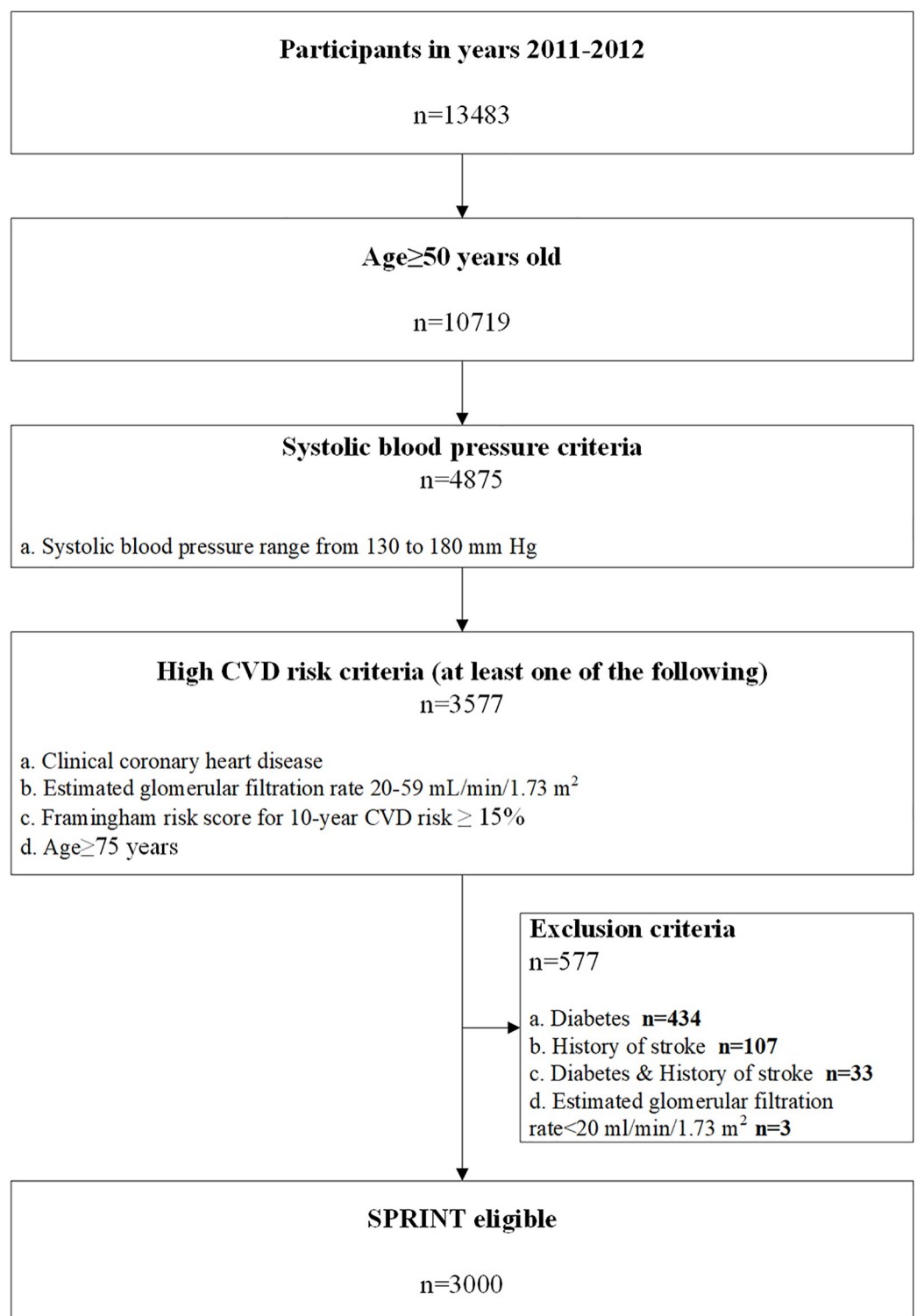

**Fig 1. Flow chart showing the eligibility criteria for SPRINT applied to the present study.** Total number of baseline participants of the China Health and Retirement Longitudinal Study was 17,708. After excluding missing blood pressure values, there were 13,483 eligible participants included in the present analysis. CVD, cardiovascular disease; SPRINT, Systolic Blood Pressure Intervention Trial.

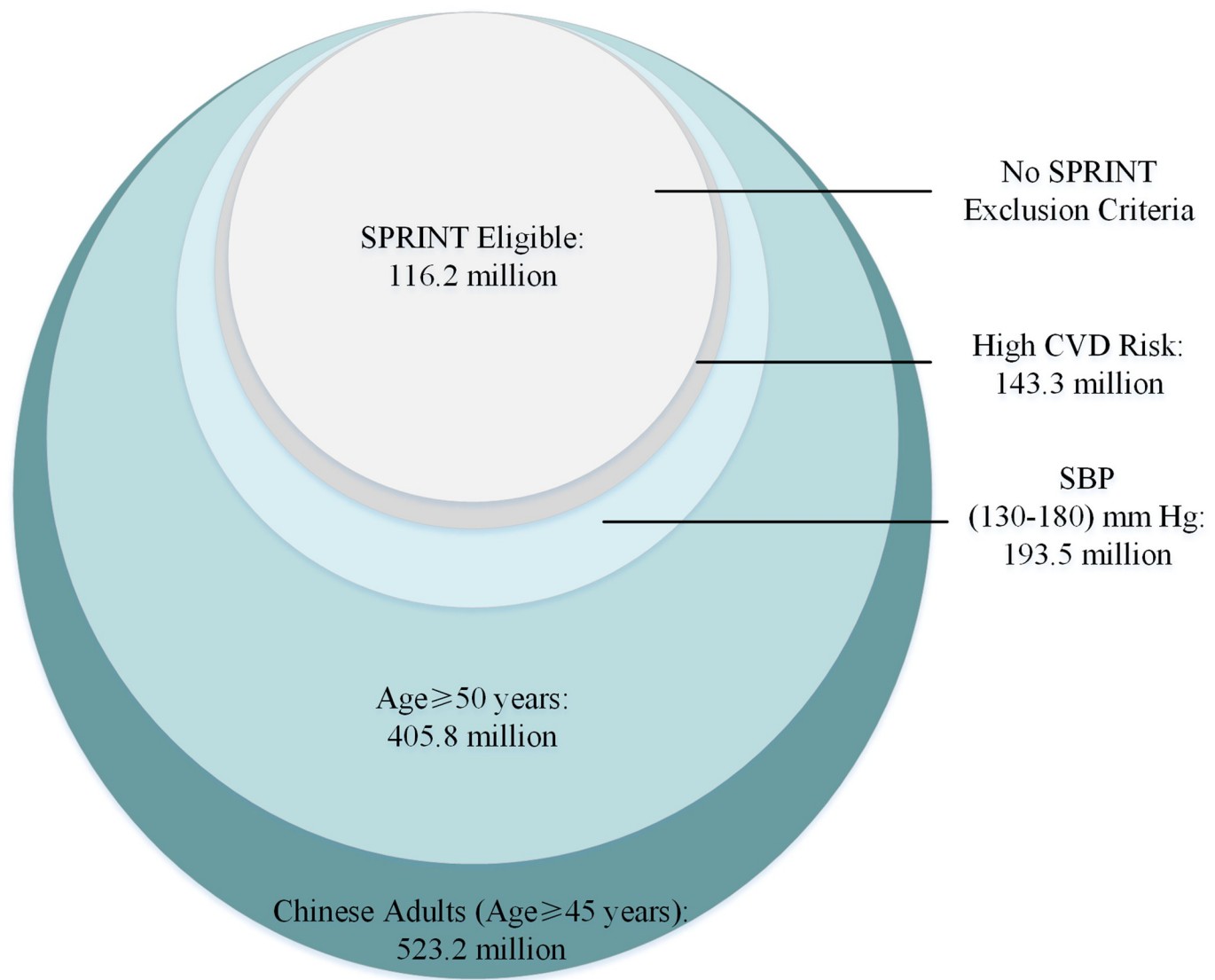

**Fig 2. Number of Chinese adults (age ≥ 45 years) meeting each sequential SPRINT eligibility criterion.** The circles are proportional to the population sizes. Hypertension defined as SBP ≥ 140 mm Hg or diastolic blood pressure ≥ 90 mm Hg or taking antihypertensive medication. CI, confidence interval; CVD, cardiovascular disease; SBP, systolic blood pressure; SPRINT, Systolic Blood Pressure Intervention Trial.

| Population | Population Number in Million | Percent (95% CI) SPRINT Eligible |
|---|---|---|
| Chinese Adults (Age≥45 years) | 523.2 | 22.2% (20.5%, 23.9%) |
| Hypertension | 225.3 | 41.4% (37.8%, 45.1%) |
| Treated Hypertension | 104.8 | 37.2% (33.6%, 40.8%) |
| Untreated Hypertension | 120.5 | 45.1% (39.0%, 51.2%) |
| Non-hypertension&SBP: 130-139 mm Hg | 60.2 | 37.9% (34.4%, 41.5%) |

$10,997 (−$752, $29,027) per additional QALY gained. The estimate of the cost-effectiveness of intensive SBP treatment in most subgroups of interest was similar to the overall estimate, although less favorable ICERs were estimated for men ($12,259 per QALY gained) and those with SBP 130–139 mm Hg ($13,277 per QALY gained).

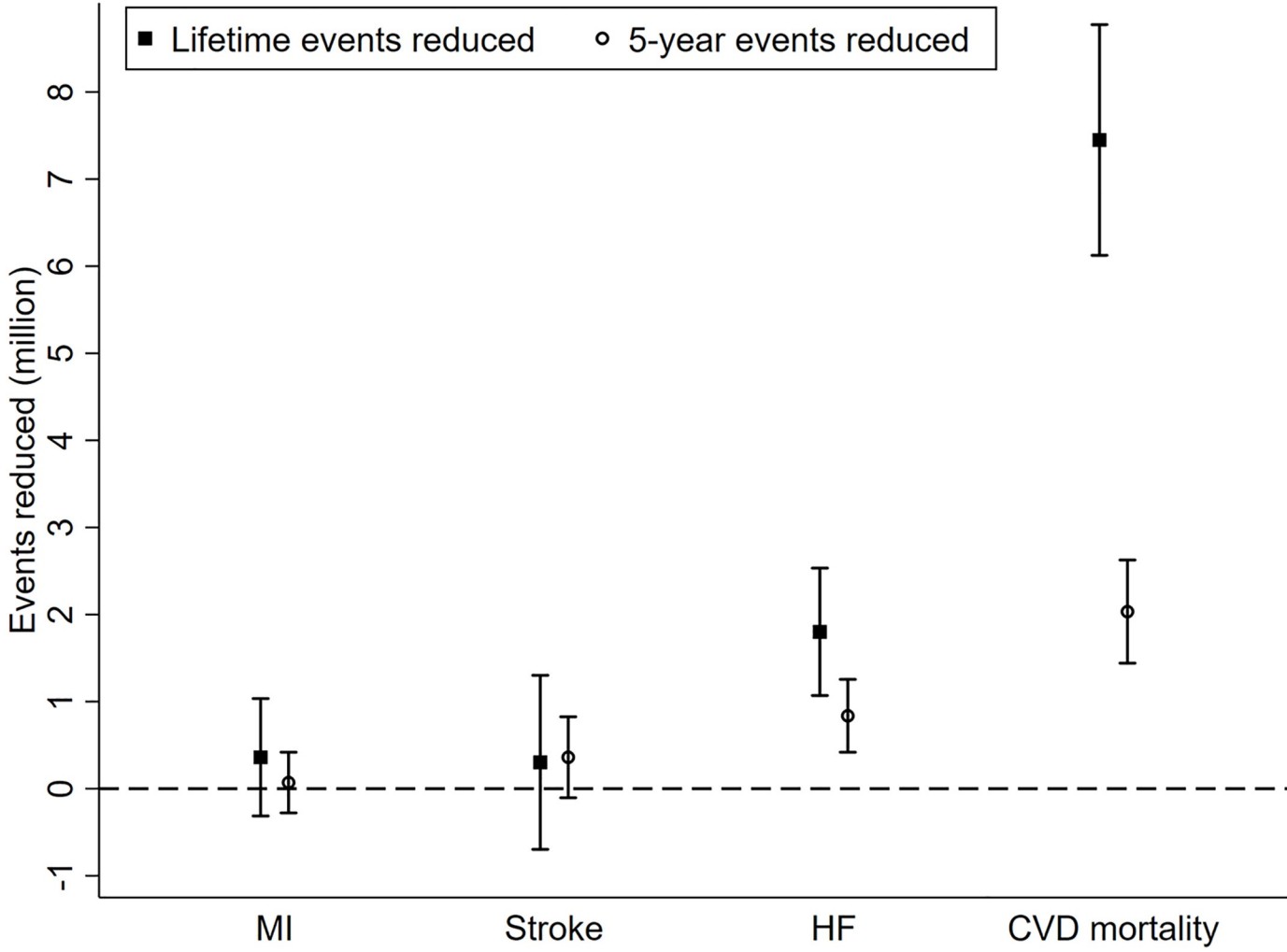

**Fig 3. Base-case 5-year and lifetime events reduced for different outcomes.** "Events reduced" refers to events that would be reduced if the intensive treatment were in place instead of standard treatment. It is calculated by multiplying the event difference from our simulation by the number of SPRINT-eligible adults. CVD, cardiovascular disease; HF, heart failure; MI, myocardial infarction.

## Uncertainty analysis

We assessed the uncertainty in the estimates by means of 1,000 probabilistic simulations. Overall, there was an 82.8% probability that intensive treatment was cost-effective at a willingness-to-pay threshold of $16,782 per QALY (1× GDP per capita in 2017) and a 98.1% probability at a threshold of $33,564 per QALY (2× GDP per capita in 2017) (Table 1; S3 and S4 Figs). The probability of being cost-effective varied across different subgroups at a threshold of 1× GDP per capita; however, it became stable (around 98% probability of being cost-effective) if the threshold increased to 2× GDP per capita.

Results in S2 Fig show that our cost-effectiveness outcome is relatively insensitive to a variety of alternate assumptions, with estimated ICERs all below 1× GDP per capita (with the exception of cost for antihypertensive medication). There were 8 parameters that were especially influential to our ICER estimates: cost for antihypertensive medication, relative risk of CVD death from treatment, relative risk of stroke from treatment, cost for SAE, cost for post-

**Table 1. Lifetime costs, QALYs, and ICER values for intensive versus standard blood pressure control strategies.**

| Outcome | Total cost, international dollars[a] | Incremental cost, international dollars (95% UI)[b] | QALYs | Incremental QALYs (95% UI)[b] | ICER (international dollars per QALY) (95% UI)[b] | Probability of being cost-effective, below 1× GDP per capita | Probability of being cost-effective, below 2× GDP per capita |
|---|---|---|---|---|---|---|---|
| Overall | | | | | | | |
| Intensive treatment | 11,395 | 3,777 (−208, 8,286) | 9.87 | 0.38 (0.13, 0.71) | 10,997 (−752, 29,027) | 0.828 | 0.981 |
| Standard treatment | 7,861 | | 9.51 | | | | |
| Male | | | | | | | |
| Intensive treatment | 11,203 | 3,806 (−188, 8,347) | 10.09 | 0.34 (0.12, 0.63) | 12,259 (−773, 33,292) | 0.769 | 0.974 |
| Standard treatment | 7,679 | | 9.74 | | | | |
| Female | | | | | | | |
| Intensive treatment | 11,723 | 3,659 (−86, 7,984) | 9.51 | 0.46 (0.16, 0.84) | 9,004 (−354, 23,427) | 0.912 | 0.988 |
| Standard treatment | 8,172 | | 9.12 | | | | |
| Age < 75 years | | | | | | | |
| Intensive treatment | 12,285 | 4,103 (−196, 9,025) | 10.75 | 0.41 (0.14, 0.77) | 11,119 (−722, 29,299) | 0.820 | 0.981 |
| Standard treatment | 8,461 | | 10.35 | | | | |
| Age ≥ 75 years | | | | | | | |
| Intensive treatment | 7,695 | 2,357 (−97, 5,056) | 6.21 | 0.25 (0.10, 0.43) | 10,176 (−373, 26,429) | 0.859 | 0.984 |
| Standard treatment | 5,365 | | 6.02 | | | | |
| SBP 130–139 mm Hg | | | | | | | |
| Intensive treatment | 11,375 | 3,914 (−178, 8,566) | 10.21 | 0.33 (0.11, 0.60) | 13,277 (−920, 33,474) | 0.729 | 0.973 |
| Standard treatment | 7,657 | | 9.87 | | | | |
| SBP ≥ 140 mm Hg | | | | | | | |
| Intensive treatment | 11,403 | 3,682 (−175, 8,076) | 9.71 | 0.42 (0.15, 0.77) | 9,916 (−598, 26,158) | 0.883 | 0.984 |
| Standard treatment | 7,957 | | 9.34 | | | | |

[a]To convert cost input to Chinese currency, one would multiply the cost by the purchasing power parity (PPP) rate (in this case, 3.54). Total cost includes antihypertension treatment costs, cost of complications, and serious adverse event costs.

[b]Probabilistic analyses based on running the model 1,000 times with the use of randomly selected values for input measurements from predefined distributions. The UIs show the 2.5 to 97.5 percentiles for the incremental differences in costs and QALYs. The uncertainty of the ICER, which was calculated as the cost per QALY gained, is shown by the probability that intensive control is cost-effective at the specified willingness-to-pay thresholds.

ICER, incremental cost-effectiveness ratio; QALY, quality-adjusted life-year; SBP, systolic blood pressure; UI, uncertainty interval.

stroke care, utility for SAE, relative risk of SAE from treatment, and relative risk of HF from treatment.

## Discussion

Using nationally representative data we found that the SPRINT intensive SBP treatment strategy could increase the number receiving SBP treatment initiation or intensification to more than 116 million adults in China, which is 22.2% of the population. Furthermore, through a validated microsimulation model we projected that over 5 years more than 2.03 million CVD deaths could be prevented, and 3.84 million life-years could be saved, if an intensive SBP treatment target of <120 mm Hg for selected high-risk patients were adopted. In addition, our study found that intensive SBP treatment would cost Int$10,997 per QALY gained and was likely to be cost-effective (82.8% probability of being below 1× GDP per capita for China in 2017).

In 2 previous studies from the US and Canada, the investigators found that 7.6% of US adults and 5.2% of Canadian adults meet the SPRINT eligibility criteria, yet our estimate for China was much higher (22.2%). These results suggest that the uncontrolled hypertension rate in China is higher than in the US and Canada. Furthermore, with such a large population, the absolute number of people who would benefit is considerably larger in China (116 million versus 16.8 million for the US and 1.3 million for Canada) [24,25]. Remarkably, we estimated that two-thirds (66.4%) of those meeting SPRINT eligibility criteria were not treated for hypertension, which was higher than the estimates for the US (50.6%) and Canada (40.4%). Additionally, we found that 22.9% of them had SBP of 130 to 139 mm Hg, and would not previously have been considered to have hypertension and therefore would not have been recommended to take antihypertensive medication under hypertension guidelines in China [19]. Correspondingly, this proportion was 19.6% in the US and 14.3% in Canada. Taken together, these findings imply that a larger population would be affected in China if intensive SBP treatment were fully implemented.

Although significant benefits from intensive SBP goals have been demonstrated in a recent meta-analysis of 34 BP-lowering trials including SPRINT [9], there is still considerable debate over the number of SAEs and the additional investments likely required, e.g., increased drug and monitoring costs [24,26]. In addressing these issues, our modeling suggests that intensive SBP treatment could reduce CVD mortality and produce gains in life-years, but with increased risk of SAEs. These findings are consistent with a report in the US population, which projected over 100,000 deaths prevented annually, but at the cost of a considerable increase in SAEs [26]. However, from a lifetime perspective, our study provides evidence to suggest that the increased costs from BP intensification and management of SAEs are offset by health gains from prevented CVD events and deaths. The ICER for intensive SBP treatment was less than 1× GDP per capita for China in 2017, with a high probability of cost-effectiveness overall (i.e., 82.8%) and in subgroups of interest (e.g., 72.9% for those with SBP between 130 and 139 mm Hg, and 91.2% for the female population). Our findings, together with those of 3 other studies with similar conclusions from the US [22,27,28], suggest that intensive SBP treatment among high-risk populations is worthy of significant capital investment, generating QALYs through reducing CVD mortality and long-term morbidity. This is also supported by our previous research, in which we found that the adoption of SBP treatment among people with SBP between 130 and 139 mm Hg could be cost-effective.

The present study has some limitations. First, not all the information required to assess eligibility for SPRINT was collected by the CHARLS survey. However, the effect of not having taken into account these certain conditions, which are expected to have a very low rate, is

thought to be minimal. Second, like all other computer simulation analyses, our approach relied on data derived from multiple sources and study types, in particular, the findings from SPRINT among the US population. Third, there is limited information on the incidence of each of the SAEs reported in SPRINT (e.g., hypotension, syncope); therefore, we could not detail the components of SAEs in our model. Also, the scarcity of data and the uncertainty of the effect of intensive management on renal complications prevented us from including long-term renal outcomes in the model. Fourth, BP measurements in SPRINT were likely done by unattended automated BP devices, which are different from the one used in CHARLS, with staff attendance during BP measurement. Studies have shown that unattended BP measurement can result in BP values that are 10–15 mm Hg lower. This may have affected the proportion of people meeting SPRINT criteria and our further modeling results. That said, evidence has shown a similar BP level and CVD risk reduction in the intensive SBP treatment group in SPRINT participants regardless of the measurement technique [29]. Fifth, our estimations of the number and proportion of Chinese adults meeting SPRINT eligibility criteria are based on survey data at 1 time point (i.e., 2011–2012), and these values may have changed with time, particularly for those with versus without hypertension treatment. Finally, although BP treatment intensification, like many other interventions, cannot not ultimately prevent the death of people, it may alter their disease progress or pathway. Therefore, our estimates of the events or deaths reduced may be confounded by the competing risk between CVD event and CVD death or non-CVD death, particularly for the lifetime estimates.

The approach described in SPRINT has had a dramatic influence on clinical practice [11,12]. The present study suggests that adoption of the SPRINT intensive SBP treatment strategy would reclassify 116 million Chinese adults as requiring SBP-lowering therapy, and if these individuals were treated, we could prevent over 2 million CVD deaths, and gain 3.84 million life-years over 5 years. Furthermore, this strategy is likely to be cost-effective over a lifetime despite greater drug costs and more SAEs.

## Supporting information

**S1 CHEERS Checklist.**
(DOCX)

**S1 Fig. Schematic depiction of the model structure.** HF, heart failure; MI, myocardial infarction.
(TIF)

**S2 Fig. One-way sensitivity analysis of model variables.** Red dashed line: 1 GDP per capita.
(TIF)

**S3 Fig. Results of the probabilistic analyses as shown in a cost-effectiveness scatter plot.**
(TIFF)

**S4 Fig. Cost-effectiveness acceptability curves.**
(TIFF)

**S1 Text. Supplementary methods and results for population and cost-effectiveness analysis.**
(DOCX)

## Author Contributions

**Conceptualization:** Chao Li, Kangyu Chen, Yang Wang, Rui Qin, Tao Chen.

**Data curation:** Xiaowei Yang, Xiaodan Yuan, Rui Qin.

**Formal analysis:** Chao Li, Zhenqiang Wu, Tao Chen.

**Investigation:** Xiaodan Yuan, Zhenqiang Wu, Duolao Wang, Tao Chen.

**Methodology:** Chao Li, Victoria Cornelius, Ewan Tomeny, Yang Wang, Xiaowei Yang, Xiaodan Yuan, Dahai Yu, Zhenqiang Wu.

**Resources:** Rui Qin.

**Supervision:** Kangyu Chen, Victoria Cornelius, Yang Wang, Dahai Yu, Duolao Wang.

**Writing – original draft:** Chao Li, Kangyu Chen, Tao Chen.

**Writing – review & editing:** Kangyu Chen, Victoria Cornelius, Ewan Tomeny, Yang Wang, Xiaodan Yuan, Rui Qin, Dahai Yu, Zhenqiang Wu, Duolao Wang, Tao Chen.

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
