## [Editor Report · Decision Letter 0]

16 Apr 2020

Dear Dr Chen, 

Thank you for submitting your manuscript entitled "Applicability and cost-effectiveness of the Systolic Blood Pressure Intervention Trial (SPRINT) in the Chinese Population" for consideration by PLOS Medicine.

Your manuscript has now been evaluated by the PLOS Medicine editorial staff and I am writing to let you know that we would like to send your submission out for external peer review.

Kind regards,

Thomas J McBride, PhD,

PLOS Medicine

---

## [Decision Letter · Decision Letter 1]

4 Jun 2020

Dear Dr. Chen,

Thank you very much for submitting your manuscript "Applicability and cost-effectiveness of the Systolic Blood Pressure Intervention Trial (SPRINT) in the Chinese Population" (PMEDICINE-D-20-01125R1) for consideration at PLOS Medicine. 

[LINK]

In light of these reviews, I am afraid that we will not be able to accept the manuscript for publication in the journal in its current form, but we would like to consider a revised version that addresses the reviewers' and editors' comments. Obviously we cannot make any decision about publication until we have seen the revised manuscript and your response, and we plan to seek re-review by one or more of the reviewers. 

We expect to receive your revised manuscript by Jun 25 2020 11:59PM. Please email us (plosmedicine@plos.org) if you have any questions or concerns.

We look forward to receiving your revised manuscript. 

Sincerely,

Emma Veitch, PhD

PLOS Medicine

On behalf of Clare Stone, PhD, Acting Chief Editor,

PLOS Medicine

plosmedicine.org

*Please revise your title according to PLOS Medicine's style - this should be structured with the main study question and then the study design (eg, : cost-effectiveness modelling) after a colon in the subtitle.

*At this stage, we ask that you include a short, non-technical Author Summary of your research to make findings accessible to a wide audience that includes both scientists and non-scientists. The Author Summary should immediately follow the Abstract in your revised manuscript. This text is subject to editorial change and should be distinct from the scientific abstract. Please see our author guidelines for more information: https://journals.plos.org/plosmedicine/s/revising-your-manuscript#loc-author-summary

*In the last sentence of the Abstract Methods and Findings section, please note any key limitation(s) of the study's methodology. The authors might also want to include a summary in the abstract of the outcomes from any of the sensitivity analyses, not just the primary analyses, in the study as this will help readers grasp how robust the findings are to underlying assumptions used in the models. 

*If possible, please modify the intext reference callouts to use numbers in square brackets (eg [1, 2]) - if referencing software is used this should be fairly quick and easy.

Comments from the reviewers:

Reviewer #1: The propsed paper is well written and with a well done statistical analysis modeling. Furthermore conclusions are driven by founded results. However some revision is needed before it could be accepted for pubblication. The mail problem is how blood pressure was measured in the study from which data were taken for the model. In fact, SPRINT trial was hardly criticized by european reserchers and guidelines since measurements were done by unattended automated BP device. If BP in the CHARLS study has been taken differently (whit classic methods) these data could not be compared with the SPRINT one. In fact, unattended BP results in 10-15 mmHg less BP and so the main criticism is that probably SPRINT is comparing actual normal control (130 mmHg) with uncontrolled BP (150 mmHg) so only confirming the actual guidelines and not confirming the intensive treatment benefit. Please better describe methods for BP measurements and discuss this issue.

Furthermore, I din't understand why in the "base-case analysis" section authors reports only on the reduction of HF incidence and CV fatal events while not considering non fatal events. Please better describe.

Reviewer #2: The manuscript concerns a very important topic of intensifying blood pressure treatment, which is a cheap and effective option that may affect a large proportion of population. However, I have a number of questions regarding the modelling assumptions:

- Which treatment effects on cardiovascular endpoints were used in the model? eTable 3 suggests these were the hazard effects coming directly from the SPRINT trial (which is exactly what I would expect). But then what is depicted in eFigure2? Also text, lines 199-201.

- Adverse events: why is the HR of 1.43 used if SPRINT reported 1.04? Further, why is a SAE assumed to incur annual costs for the lifetime duration but the QoL decrement is assumed to be back to normal after one week? Where would the costs come from then?

- Why is it assumed that further CV events may only happen within the next year of the first one?

- In terms of the proportion of people not treated for hypertension, is it possible to use re-interview data, as presumably the situation has changed wince 2011.

- eTable 5: could the authors add description of the population / any other assumptions used in the comparator table, to make it clearer that like-with-like is being compared.

- One ICER may mask important differences in population subgroups. Could the results be presented by age groups and categories of blood pressure.

I am also struggling to understand information presented in Figure 3. What scale is it presented on? The x-axis suggests it may be on a person-level, but how can 7 deaths be averted for one person? In general, information on lifetime number of deaths avoided is not necessarily meaningful - people will have to die eventually; besides those that live longer, may well experience CV events just as much as those who live less; it's just they experience them later. Perhaps something like "numbers needed to treat to avoid 1 stroke in 5 years" would be more meaningful. 

Related to this, in eTable 6, the numbers of strokes saved over lifetime is smaller than that saved over 5-years, with the trend reversed with every other endpoint. There may well be a reason for this, but it should be noted and discussed.

A few minor comments:

- Table 1: costs and ICERs should be rounded to the nearest integer, if not more; also it should be clarified what "incremental costs" refers to - total costs or just hospital costs? Ir the former, it would be very helpful to see specifically hospital costs. Treatment costs could be deduced from life expectancy & treatment cost.

- There were quite a few spelling errors, particularly in the appendix, could this be reviewed carefully.

Reviewer #3: This is a statistical review of manuscript PMEDICINE-D-20-01125R1. It is well structured, well written and the topic is of great importance. From a statistical viewpoint, I would like to see 2 main clarifications: 

1/ Line 236 : Our model showed a reasonable concordance between model predicted risk estimates and published hazard ratios (HRs) (0.86 vs.0.83 for MI, 0.95 vs. 0.89 for stroke, 0.65 vs. 0.62 for HF, 0.59 vs. 0.57 for CVD mortality, 0.85 vs. 0.73 for all-cause mortality). What are these HRs ? Are they the ratios of the hazards of MI, Stroke etc between patients on intensive treatment vs standard? 

Figure S2 should explain in the legend what HRs are plotted

2/ Figure 3: what is the scale of the Y-axis? Is it at the level on an individual, or is it at the scale of the whole Chinese population (based on the text line 251 this is probably the case)? Some CIs reach the negative territory - does it mean that the intensive treatment may increase MI and stroke? 

I think that the legend should make clear that these are the events that would be avoided if the intensive treatment was in place instead of the standard one. 

Minor points: in some instances, there are sentences where words are missing or the English could be improved. For example: 

"Although adoption of the SPRINT treatment strategy would increase the number of Chinese adults requiring SBP [treatment?] intensification" 

"this approach have [has?] the"

"the detailed disease states within each path could be [can be found?] in our previous publication"

[LINK]

---

## [Decision Letter · Decision Letter 2]

27 Aug 2020

Dear Dr. Chen,

Thank you very much for submitting your revised manuscript "Applicability and Cost-effectiveness of the Systolic Blood Pressure Intervention Trial (SPRINT) in the Chinese Population: Cost-effectiveness Modelling" (PMEDICINE-D-20-01125R2) for consideration at PLOS Medicine. 

Your paper was evaluated by a senior editor and discussed among all the editors here. It was also discussed with an academic editor with relevant expertise, and sent to one of the original reviewers. The review is appended at the bottom of this email and any accompanying reviewer attachments can be seen via the link below:

[LINK]

In light of these reviews, I am afraid that we still will not be able to accept the manuscript for publication in the journal in its current form, but we would like to consider a revised version that addresses the reviewers' and editors' comments. Obviously we cannot make any decision about publication until we have seen the revised manuscript and your response, and we plan to seek re-review by one or more of the reviewers. 

We expect to receive your revised manuscript by Sep 17 2020 11:59PM. Please email us (plosmedicine@plos.org) if you have any questions or concerns.

We look forward to receiving your revised manuscript. 

Sincerely,

Thomas McBride, PhD

Senior Editor 

PLOS Medicine

plosmedicine.org

1- Thank you for editing your title. Please edit again slightly: “Applicability and Cost-effectiveness of the Systolic Blood Pressure Intervention Trial (SPRINT) in the Chinese Population: a Cost-effectiveness modelling study”

2- Thank you for adding an Author Summary. Please edit the last point of the Author Summary to: “This evidence suggests that intensive treatment to reducing blood pressure among high-risk populations could be cost effective.” (or something similar).

3- Throughout the manuscript, please include the p-values alongside 95% CIs when relevant.

4- Thank you for adding the non-fatal events in response to reviewer 1’s second point. It does look like for MI and stroke, the reductions are not significant. Please make this clear by separating these outcomes out in the text.

5- Please include the url and any accession numbers necessary to apply for access to the CHARLS dataset.

6- Did your study have a prospective protocol or analysis plan? Please state this (either way) early in the Methods section.

Comments from the reviewers:

Reviewer #2: Thank you for providing answers to my comments. I would still like to have a few points (from my original comments) clarified, and it would be helpful if these were clarified in the manuscript text as well.

* Treatment effect. Thank you for clarifying these were the SPRINT treatment effects that were used in the modelling. I would then suggest removing S2 Figure from the manuscript as it does not add anything. The simulated HRs are not simulated, but are taken from literature, so of course they will be the same when plotted against the HRs they are equal to. The small difference presumably comes from projecting subsequent events, population sampling and randomness in the simulation process.

* Cost of SAEs. So if an SAE has occurred, will the costs be applied every single year after it occurred (despite QoL being back no normal within a week) or only in a year that it occurred? 

* Modelling subsequent CV events. I understand that secondary events may happen, but why can they only happen in the year after the primary event? Whilst the risk of a subsequent event is higher in the next year, it does not go away in years after that

* Proportion of people on hypertension. This comment referred to the authors' estimation that 2/3 of those meeting SPRINT eligibility criteria were not treated for hypertension (line 336, page 15; also abstract). However, this was the case in 2011, since then most likely this number has decreased. I assume that in the "standard treatment" scenario everyone was prescribed antihypertensive medications rather than just those that were recorded to do so. Could the authors confirm this please.

* eTable5: The added text just re-phrases the results that are presented. it still not clear that the populations for which results are presented (ie population in this study vs populations in references 30-37) are comparable. Could this be commented on please.

* Figure 3. What exactly does it mean to say that 7 million CVDs were averted over lifetime? Does it mean that these 7 million people died of non-CVD causes instead? It seems misleading to present this number unless there is some discussion in the text.

* Comment on Table 1: It is still unclear what costs are presented in column named "Costs, Int $". Are these hospital costs, or do they include treatment costs? Regardless, it would be very helpful to look separately at hospital costs and treatment costs. 

* Spelling: there are still some spelling mistakes (eg Figure 2: "adverted" should be "averted")

[LINK]

---

## [Decision Letter · Decision Letter 3]

9 Dec 2020

Dear Dr. Chen,

Thank you very much for re-submitting your manuscript "Applicability and Cost-effectiveness of the Systolic Blood Pressure Intervention Trial (SPRINT) in the Chinese Population: a cost-effectiveness modelling study" (PMEDICINE-D-20-01125R3) for review by PLOS Medicine.

I have discussed the paper with my colleagues and it was also seen again by one reviewer. 

[LINK]

We look forward to receiving the revised manuscript by Dec 16 2020 11:59PM.   

Sincerely,

Thomas McBride, PhD

Senior Editor 

PLOS Medicine

plosmedicine.org

Requests from Editors:

1- Please address the remaining comments from reviewer 2 and incorporate your responses into the main text (not just the rebuttal letter), in case readers are also unclear.

2- Please report your economic analysis according to the appropriate study design provided at http://www.equator-network.org/?post_type=eq_guidelines&eq_guidelines_study_design=economic-evaluations&eq_guidelines_clinical_specialty=0&eq_guidelines_report_section=0&s= and provide the relevant completed checklist. In the checklist please include sufficient text excerpted from the manuscript to explain how you accomplished all applicable items.

Please add the following statement, or similar, to the Methods: "This study is reported as per the XXX guideline (S1 Checklist)."

3- Thank you for updating your data statement. However, I note that the webpage you direct to reads: “All data will be made public one year after the end of data collection.” PLOS requires that the data underlying all analyses in the paper be available at the time of publication. Can you confirm that the data are available now? If so, update the data statement to read: “CHARLS data are available via the study website: ” and provide a more specific url to direct readers to the dataset. If the dataset is not currently available, please let us know the date when it will be available.

It seems that this paper uses data from wave 1 and 2 (2011-2012), which seem to be available 

here:

http://charls.pku.edu.cn/pages/data/2011-charls-wave1/en.html

and here:

http://charls.pku.edu.cn/pages/data/2012-charls-pilot-wave2/en.html

If that is correct, please use these urls in the data statement.

4- Line 65: remove “horizon”: “We estimated that over five years…”

5- Thank you for editing the language around MI and stroke. However, please further edit the line in the abstract to reflect that the 95% CIs do not predict a reduction in MI or stroke incidence. I suggest moving these outcomes to a separate sentence, e.g., “Estimated reductions of 0.069 million (95% CI: -0.28, 0.42) myocardial infarction and 0.36 million (95% CI: -0.10, 0.82) stroke incidences were not statistically significant.”

6- Thank you for your response regarding the planned analyses and your previous published model. Please note this in the methods section and cite the study.

7- At lines 56 and 157, “... aged over 45 *years*...”

8- Line 77 and throughout, please replace “subjects” with “people”.

9- Throughout the manuscript, please replace the abbreviation “y” with “years”.

10- Please remove “As a landmark study in hypertension management,” from line 388.

11- Please include information on ethics approval and consent obtained for the CHARLS study when mentioning it in the Methods section.

12- The supporting information, please replace “gender” with “sex”.

Comments from Reviewers:

Reviewer #2: 

please see attached file

[LINK]

---

## [Editor Report · Decision Letter 4]

22 Dec 2020

Dear Dr. Chen,

Thank you very much for re-submitting your manuscript "Applicability and Cost-effectiveness of the Systolic Blood Pressure Intervention Trial (SPRINT) in the Chinese Population: a cost-effectiveness modelling study" (PMEDICINE-D-20-01125R4) for review by PLOS Medicine.

After reviewing your most recent revision, there are 2 remaining editorial points to be addressed before we can accept.

The remaining issues that need to be addressed are listed at the end of this email. Please take these into account before resubmitting your manuscript:

[LINK]

We expect to receive your revised manuscript within 1 week, but I understand that I am sending this during the holiday season. Please know that the due dates listed here are flexible. Please email us (plosmedicine@plos.org) if you have any questions or concerns.

We look forward to receiving the revised manuscript by Dec 29 2020 11:59PM.   

Sincerely,

Thomas McBride, PhD

Senior Editor 

PLOS Medicine

plosmedicine.org

Requests from Editors:

1- Thank you for clarifying the data availability. Please also update the data availability statement in the submission metadata to read:

“CHARLS data are available via the website: http://charls.pku.edu.cn/pages/data/2011-charls-wave1/en.html

Other relevant data are within the manuscript and its Supporting Information files”

2- In your response to Reviewer 3, comment 5, you change “events averted” to “events reduced”, but I’m not sure that clarifies what the figure is reporting. Please add a bit more detail to the description in the figure legend, so it’s clear what outcomes are represented in the figure.

[LINK]

---

## [Editor Report · Decision Letter 5]

2 Feb 2021

Dear Dr Chen, 

On behalf of my colleagues and the Academic Editor, [Kazem Rahimi], I am pleased to inform you that we have agreed to publish your manuscript "Applicability and Cost-effectiveness of the Systolic Blood Pressure Intervention Trial (SPRINT) in the Chinese Population: a cost-effectiveness modelling study" (PMEDICINE-D-20-01125R5) in PLOS Medicine.

PRESS

Sincerely, 

Raffaella

Dr Raffaella Bosurgi MSc, PhD

Executive Editor, PLOS Medicine

rbosurgi@plos.org

https://twitter.com/raffi74

Remote based in London, UK

PLOS
